# PRIVATE AND INTERPRETABLE CLINICAL PREDICTION WITH QUANTUM-INSPIRED TENSOR TRAIN MODELS

## ABSTRACT

Machine learning in clinical settings must balance predictive accuracy, interpretability, and privacy. Models such as logistic regression (LR) offer transparency, while neural networks (NNs) provide greater predictive power; yet both remain vulnerable to privacy attacks. We empirically assess these risks by designing attacks that identify which public datasets were used to train a model under varying levels of adversarial access, applying them to LORIS, a publicly available LR model for immunotherapy response prediction, as well as to additional shallow NN models trained for the same task. Our results show that both models leak significant training-set information, with LRs proving particularly vulnerable in white-box scenarios. Moreover, we observe that common practices such as cross-validation in LRs exacerbate these risks. To mitigate these vulnerabilities, we propose a quantum-inspired defense based on tensorizing discretized models into tensor trains (TTs), which fully obfuscates parameters while preserving accuracy, reducing white-box attacks to random guessing and degrading black-box attacks comparably to Differential Privacy. TT models retain LR interpretability and extend it through efficient computation of marginal and conditional distributions, while also enabling this higher level of interpretability for NNs. Our results demonstrate that tensorization is widely applicable and establishes a practical foundation for private, interpretable, and effective clinical prediction.

## 1 INTRODUCTION

Machine learning (ML) is increasingly used for clinical prediction but poses critical privacy risks, as models trained on sensitive medical data can inadvertently leak individual information (Fredrikson et al., 2014; Sweeney, 2015). In domains where interpretability is essential, intuitive models like logistic regression (LR) are often preferred, yet they are particularly vulnerable to such attacks. More complex models like neural networks (NNs) are harder to attack, but their complexity also makes it challenging to design strong, accuracy-preserving defenses, leaving them vulnerable.

In this work, we propose a quantum-inspired approach to privacy protection based on tensor network (TN) models, focusing on tensor trains (TT). Building on recent methods for tensorizing pre-trained ML models into TT form (Pareja Monturiol et al., 2025), we tensorize clinical models to enhance privacy while preserving accuracy and interpretability. While the TT form has already been shown to offer strong white-box (WB) privacy (Pozas-Kerstjens et al., 2024), we further enhance black-box (BB) privacy by integrating discretized output scores into the tensorization process.

To assess the privacy risks of clinical models and the protection offered by TTs, we attack LORIS (Chang et al., 2024), a publicly available LR model for immunotherapy response prediction hosted on a U.S. government website. We design a membership inference attack under both BB and WB access, using a shadow model approach that trains multiple models with varied hyperparameters and datasets, followed by an adversarial meta-classifier to predict which public datasets were included in the training set. To further demonstrate the generality of our tensorization approach, we perform analogous experiments on shallow NNs trained with the same data and objectives. Additionally, we compare the results with Differential Privacy (DP) defenses for both LR and NN models.

Our results show that tensorizing the models degrades attack performance across all access levels, reducing WB attacks to random guessing, while providing BB protection comparable to DP

and maintaining predictive accuracy close to the unprotected models. We also find that the size of the discretization steps for the output scores provides control over privacy protection, in a manner analogous to how DP is tuned by adding calibrated noise (Dwork, 2006a). Additionally, we show that common practices like cross-validation, when used to deploy averaged models as in LORIS, can severely compromise privacy, enabling accurate training-set identification even from BB access via the public web interface. TT approximations preserve key properties of LORIS, such as response monotonicity, while enhancing interpretability through efficient computation of marginals and conditionals. This supports feature-sensitivity analysis and enables the construction of cancer-type-specific models without retraining. Importantly, the same techniques can be directly applied to tensorized NNs, providing interpretability for these black-box models.

Although we propose discretizing output scores as a heuristic to *compress* the output space and hinder membership inference, the tensorization mechanism itself is independent of the obfuscation method. Alternative approaches, such as those in Jia et al. (2019); Yang et al. (2020) or DP-based methods like in Ye et al. (2022), could be applied similarly before tensorizing. Hence, we highlight the generality of the tensorization process, which can be applied post-training as a practical strategy for privacy-preserving, interpretable, and effective models—crucial features in sensitive domains such as clinical prediction, where we advocate for its routine use.

The remainder of this paper is structured as follows. Section 2 reviews related work and preliminaries. Section 3 outlines our setting, attack, and defenses, and presents the results. Section 4 analyzes the interpretability of TT models in comparison to LORIS. Finally, Section 5 discusses conclusions and future directions.

## 2 RELATED WORK AND PRELIMINARIES

The widespread adoption of ML systems increases the risk of leaking sensitive personal data. Prior work has extensively examined these vulnerabilities and proposed various defenses.

### 2.1 PRIVACY ATTACKS

A wide range of attacks exploit privacy vulnerabilities in ML, leveraging either BB or WB access. Key examples include model inversion (Fredrikson et al., 2014), model classification (Ateniese et al., 2015), and membership inference (Shokri et al., 2017), which vary in scope from extracting individual samples to uncovering global patterns. In this work, we adopt the membership inference approach to identify groups of samples present in the training set.

More recently, reconstruction attacks have aimed to recover exact training samples. Some rely on shadow-model training (Balle et al., 2022), while others exploit optimization properties of models trained with Stochastic Gradient Descent (SGD) (Haim et al., 2022; Oz et al., 2024). Notably, for LR, such attacks can yield closed-form solutions (Balle et al., 2022), underscoring the vulnerability of simple, widely used models.

### 2.2 DEFENSE MECHANISMS

Given the diversity of privacy-related attacks, various defense mechanisms have been proposed. Among these, Differential Privacy (Dwork, 2006b) stands out for its rigorous framework. DP quantifies the likelihood that an attacker can infer whether a specific user's data was included in a statistical process. A randomized algorithm $\mathcal{A}$ is $\varepsilon$-DP if, for any set of outcomes $\mathcal{S}$ in the range of $\mathcal{A}$, it satisfies:

$$\log\left(\frac{P[\mathcal{A}(D) \in \mathcal{S}]}{P[\mathcal{A}(D') \in \mathcal{S}]}\right) \leq \varepsilon, \tag{1}$$

where $D$ and $D'$ differ by a single element. This metric guides the addition of calibrated noise to achieve a target $\varepsilon$, based on the sensitivity of the function being protected (Dwork, 2006a; Dwork & Roth, 2014). For LR models, common defenses add noise either to the objective or to the final parameters (Chaudhuri et al., 2011), whereas the standard NN defense, DP-SGD (Abadi et al., 2016), introduces noise into the gradients at each training step. However, the noise required for strong privacy guarantees often degrades performance and may exacerbate group disparities (Bagdasaryan et al., 2019; Hansen et al., 2024). As a result, there is no consensus on how to set $\varepsilon$ meaningfully

(Garfinkel et al., 2018); while small values are theoretically ideal, larger values may still prevent reconstruction attacks in practice without significantly harming accuracy (Ziller et al., 2024).

Beyond DP, recent work has explored whether standard ML practices can improve privacy. Pruning introduces small errors that resemble DP-like protection (Huang et al., 2020), while knowledge transfer reduces dependence on specific training data (Shejwalkar & Houmansadr, 2020). Several works show that non-private models produce overly spread output scores, and that compressing this space, e.g., by injecting crafted noise to prevent membership inference, improves privacy (Jia et al., 2019; Yang et al., 2020; 2023). With a similar goal, other approaches add noise to the output scores, providing DP guarantees with minimal utility loss (Ye et al., 2022; Papernot et al., 2017).

Our approach builds on these ideas: tensorization acts as a knowledge-distillation mechanism that converts any model into an efficient, interpretable representation that preserves WB privacy (Pozas-Kerstjens et al., 2024). To further enhance BB privacy, we apply tensorization to discretized scores that collapse the model's output space into a smaller subdomain—though other output-obfuscation methods could similarly be applied to enforce DP. Thus, our approach may resemble work on private low-rank approximation, such as Kapralov & Talwar (2013), but at the level of full-model decomposition, in contrast with techniques which use low-rankness solely for private fine-tuning (Liu et al., 2025).

## 2.3 Tensor train models

Tensor networks are low-rank decompositions of high-dimensional tensors with roots in quantum many-body physics. They offer compact, interpretable representations of quantum states (Pérez-García et al., 2007; Orús, 2014; Cirac et al., 2021) and have recently been adapted to machine learning (Stoudenmire & Schwab, 2016; Novikov et al., 2018). TNs have been applied to model compression (Novikov et al., 2015; Tomut et al., 2024), explainable AI (Tangpanitanon et al., 2022; Aizpurua et al., 2024), anomaly detection (Wang et al., 2020), and robustness (Mossi et al., 2025). Importantly, TNs offer formal WB privacy guarantees: multiple parameterizations can represent the same model, effectively obfuscating all but its BB behavior (Pozas-Kerstjens et al., 2024).

Throughout this work, we focus on one-dimensional TNs known as tensor trains (Oseledets, 2011). An order-$N$ tensor $T \in \mathbb{R}^{d^N}$ admits a TT representation with *ranks* $r_n$ if it can be written as

$$T(i_1, \ldots, i_N) = G_1(i_1) \cdots G_N(i_N), \tag{2}$$

where the *cores* $G_n$ are $r_{n-1} \times r_n$ matrices and $r_0 = r_N = 1$. This structure also supports continuous functions of the form

$$f(x_1, \ldots, x_N) = \sum_{i_1, \ldots, i_N} \mathrm{W}(i_1, \ldots, i_N) \, \phi_1(i_1, x_1) \cdots \phi_N(i_N, x_N), \tag{3}$$

where $\mathrm{W}$ is a TT-format coefficient tensor and $\phi_n(i_n, x_n)$ are vector-valued *embedding* functions indexed by $i_n$. To ensure non-negative probability scores, it is standard to define distributions via the Born rule: $p(x) = |f(x)|^2$. Further details on TTs, including efficient marginalization and conditioning, are provided in Appendix A.

TTs can be trained using SGD or physics-inspired variants (Stoudenmire & Schwab, 2016). Alternatively, TT representations can be constructed via low-rank decompositions, bypassing high-dimensional optimization. Recent techniques based on sketching (Hur et al., 2023) and cross interpolation (Fernández et al., 2025) achieve this using only function evaluations—i.e., BB access—to approximate continuous functions in TT form. A recent method, TT-RSS, extends this idea to tensorize pre-trained NNs using a small evaluation dataset (Pareja Monturiol et al., 2025). This results in an efficient procedure, requiring $O(|D|^2 Nd)$ model evaluations on a set of *pivots* $D$ and an additional $O(|D|^3 Nd)$ to assemble the TT. In this work, we adopt TT-RSS to tensorize models.

## 3 Privacy Analysis

To evaluate the privacy risks of clinical prediction models and compare defense strategies, we design a membership inference attack based on shadow-model training. Assuming an adversary with access to multiple public datasets, the attack determines which of them were used to train a model under

varying levels of access. Although this task is easier than individual membership inference, in medical settings each public dataset may correspond to a small patient cohort, whose identification can still reveal sensitive information. As a case study, we target LORIS, a publicly available LR model introduced by Chang et al. (2024) for immunotherapy response prediction. We additionally consider NN models for the same task. Below we define the setting, describe the adversarial assumptions and attack, and present the experimental setup, with results reported at the end of the section.

### 3.1 SETTING AND NOTATION

Let $\mathcal{D} = \{D_1, \ldots, D_M\}$ be the set of public datasets, and define $\mathcal{D}_\cup = \{\bigcup \mathcal{C} \mid \mathcal{C} \in \mathcal{P}(\mathcal{D}) \setminus \{\varnothing\}\}$, where $\mathcal{P}(\mathcal{D})$ is the power set. A training set $D_\cup \in \mathcal{D}_\cup$ is the union of one or more $D_m \in \mathcal{D}$. Using the indicator vector $\mathbf{1}(D_\cup)$, we represent $D_\cup$ as a multi-hot vector with entries 1 for datasets $D_m \subset D_\cup$ and 0 otherwise.

We define the training algorithm as follows: given a model architecture $\Phi$, hyperparameters $H_\Phi \in \mathcal{H}_\Phi$, and a training set $D_\cup \in \mathcal{D}_\cup$, the training mechanism $\mathcal{T}_\Phi : \mathcal{H}_\Phi \times \mathcal{D}_\cup \to \Theta$ outputs parameters $\theta \in \Theta$ such that $\Phi_\theta(\cdot)$ is a trained model. In practice, $\mathcal{T}_\Phi$ is stochastic due to factors such as random initialization or mini-batch selection in SGD, so for fixed $H_\Phi$ and $D_\cup$ we interpret $\mathcal{T}_\Phi(H_\Phi, D_\cup)$ as sampling from a model distribution. In addition, since training data are typically standardized for stability, yielding coefficients defined on standardized inputs, we assume that $\mathcal{T}_\Phi$ returns rescaled parameters that operate on raw input data. Details of the standardization and rescaling procedures are provided in Appendix B.

To mitigate bias and overfitting, it is standard to use $K$-fold cross-validation, which partitions $D_\cup$ into $K$ folds and trains $K$ models, each on $K - 1$ folds. We denote by $\mathcal{T}_\Phi^{J,K} : \mathcal{H}_\Phi \times \mathcal{D}_\cup \to \Theta$ the procedure that applies $\mathcal{T}_\Phi$ with fixed $H_\Phi$, performs $J$ repetitions of $K$-fold cross-validation, corrects for feature standardization, and averages the resulting parameters into a final model.

### 3.2 DESCRIPTION OF THE ATTACK

The adversary knows the model architecture $\Phi$, the public datasets $\mathcal{D}$, and a finite set of hyperparameters $\mathcal{H}_\Phi$. They also know the training mechanisms $\mathcal{T}_\Phi, \mathcal{T}_\Phi^{J,K}$, and have sufficient resources to train shadow models and meta-classifiers. Access to the target model is limited to restricted information $h(\Phi_\theta)$, which we categorize into three independent access levels:

- $b$-Weak black-box ($b$-WBB): Access to outputs discretized into $b$ bins, e.g., $b = 2$ gives binary outputs. Values $< 0.5$ map to the lower bin limit, and $> 0.5$ to the upper.

- Strong black-box (SBB): Access to raw continuous scores. As $b$ grows to machine precision, $b$-WBB converges to SBB.

- White-box (WB): Access to model parameters. Although parameters allow computing outputs, we treat BB and WB separately to assess each source of information, while stronger attacks may combine both.

The attack proceeds by constructing a dataset of shadow models, each trained under different hyperparameter configurations and training sets. From each model, we collect the relevant information together with the corresponding public datasets used for training. This forms the input to a multi-label classifier, which learns to identify the presence of public datasets in the training sets. Formally, the attack consists of the following steps:

1. For each $H_\Phi \in \mathcal{H}_\Phi$ and $D_\cup \in \mathcal{D}_\cup$, train $R$ shadow models using $\mathcal{T}_\Phi$ or $\mathcal{T}_\Phi^{J,K}$.

2. Build $\{(h(\Phi_{\theta^i}), \mathbf{1}(D_\cup^i))\}_{i=1}^{R|\mathcal{H}_\Phi||\mathcal{D}_\cup|}$, where $h(\cdot)$ denotes available information: under BB access it returns outputs on $S$ samples, and under WB access it returns parameters $\theta^i$.

3. Train an adversarial model minimizing independent cross-entropy losses for each $D_m$, yielding $\mathcal{A} : \Theta \to [0, 1]^M$, where entry $m$ gives the probability that $D_m \subset D_\cup$.

### 3.3 EXPERIMENTAL SETUP

We briefly describe the datasets, models, and implementation details. All experiments[1] were run on an Intel Xeon CPU E5-2620 v4 with 256 GB RAM and an NVIDIA GeForce RTX 3090, using Scikit-Learn for LR models and NN-based attacks (Pedregosa et al., 2011), Diffprivlib for DP variants (Dwork, 2006b), PyTorch for NN models (Paszke et al., 2019), Opacus for DP-SGD training (Yousefpour et al., 2022), and TensorKrowch for TT models (Pareja Monturiol et al., 2024).

#### 3.3.1 DATASETS

To build the public set $\mathcal{D}$ we use the cohorts employed to train and evaluate LORIS, which include clinical, pathological, and genomic features with a binary treatment-response label. For details see Appendix C or Chang et al. (2024); we list them here with shorthand identifiers and sample sizes: Cho1 (964) and Cho2 (515), *train* and *test* partitions from Chowell et al. (2022); MSK1 (453) and MSK2 (104) from Chang et al. (2024); Shim (198) from Shim et al. (2020); and Kato (35) from Kato et al. (2020). In all cases, response is imbalanced, with ∼30% of patients responding to treatment.

We use 6-feature models: Tumor Mutational Burden (TMB), Previous Systematic Therapy History (PSTH), Albumin, Neutrophil-to-Lymphocyte Ratio (NLR), Age, and Cancer Type. Cancer Type is divided into 16 binary variables, yielding 21 input features in total.

#### 3.3.2 TARGET MODELS

As target models, we consider LRs and NNs. Following Chang et al. (2024), we train *averaged* LRs via $\mathcal{T}_\Phi^{J,K}$ with $J = 20$ and $K = 3$. While LORIS used larger values of $J$ and $K$, we found this configuration sufficient to obtain comparable results. For comparison, we also train *vanilla* LRs through a single run of $\mathcal{T}_\Phi$ on an 80% split of $D_\cup$. In both cases, the hyperparameters are solver = "saga", penalty = "elasticnet", class_weight = "balanced", max_iter = 100, l1_ratio $\in \{0, 0.5, 1\}$, and C $\in \{0.1, 1, 10\}$, forming the uncertainty set $\mathcal{H}_\Phi$. For NNs, we train 2-layer MLP classifiers following the same procedure as the vanilla LRs. We adopt the best hyperparameters reported by Chang et al. (2024): two hidden layers of size 19, binary cross-entropy loss, and Adam optimization for 100 epochs with batch_size = 32, lr = $10^{-3}$ and weight_decay = $10^{-5}$.

For each dataset $D_\cup$, hyperparameter configuration $H_\Phi$, and training method (vanilla or averaged), the adversary trains $R = 100$ models. Each model is then tensorized via TT-RSS (Pareja Monturiol et al., 2025), using 50 random samples from $D_\cup$ as pivots, evaluated under $b$-WBB access to the original model, with $b \in \{2, 6, 10\}$. The resulting TTs have $N = 22$ cores (including one for the output), ranks $r_n = 2$ for all $n$, input dimensions $d = 2$, and use polynomial embeddings $\phi_n(\cdot, x) = [1, x]$. To accommodate the higher complexity of NNs, we tensorize them using 80 pivots and ranks $r_n = 5$. After tensorization, the TT cores are randomized via a gauge transformation to prevent any leakage under WB access, fully obfuscating the TT parameters.

Due to the monotonicity of LR, model parameters can be exactly recovered from scores (see Appendix D), making SBB and WB access equivalent, although WB is typically easier to exploit. Since tensorization approximates LR outputs with a TT representation, it is also possible to recover LR coefficients from TT evaluations. Thus, because TT coefficients are fully obfuscated, we assume a WB attacker would instead reconstruct the original LR coefficients and attack those directly; accordingly, we report only these attacks in Table 1. This reconstruction is not possible for tensorized NNs, for which the attacker only has access to the TT parameters.

Finally, for comparison with a standard privatization approach, we also train DP models (LR-DP, NN-DP) from scratch. Since DP training of LR is restricted to solver = "lbfgs" and penalty = "l2", we fix max_iter = 100 and vary the privacy budget $\varepsilon \in \{0.1, 1, 10, 100\}$, where $\varepsilon = 100$ nearly matches the non-DP case. Only vanilla models are considered, as averaging would cancel the injected noise and effectively increase $\varepsilon$. For NNs, we follow the same training setup as in the non-DP case but apply DP-SGD with max_grad_norm = 1, $\delta = 10^{-4}$, and $\sigma \in \{20, 5, 1, 0\}$, which correspond approximately to privacy budgets $\varepsilon \in \{0.2, 1, 10, \infty\}$. To achieve these budgets, we reduce the number of epochs to 50.

---

[1]The code is publicly available at: https://anonymous.4open.science/r/tts4privacy

### 3.3.3 ADVERSARIAL MODELS

To attack the models described above we use NN-based adversaries: MLP multi-label classifiers with three hidden layers of sizes 32, 16 and 8, and an output layer of size 6 (one output per public dataset). The input layer size depends on the access type. For BB attacks, each shadow model is evaluated on $S = 100$ samples (the same $S$ samples for all models), randomly drawn from $\bigcup \mathcal{D}$; the resulting vector of raw or discretized outputs is the adversary input. For WB attacks we collect full model parameters: for LR, 22 parameters (21 coefficients + intercept); for NN, all per-layer parameters concatenated into an 818-dimensional vector; and for TT, all $N = 22$ cores concatenated into a single vector of 168 (TT-LR) or 1 020 (TT-NN) dimensions. All parameters are rescaled when needed to operate on raw inputs (see Appendix B). The MLPs are trained with activation = "relu", solver = "adam" and max_iter = 100. Since WB attacks exhibited greater variability, we applied 5-fold cross-validation with predictions averaged across folds; on top of this, to obtain robust statistics we repeat 5-fold cross-validation five times for both WB and BB attacks.

### 3.4 RESULTS

To evaluate the overall performance of our attacks, Table 1 reports Hamming scores, i.e., the proportion of correct label predictions across all public datasets and shadow-model instances, akin to the attack success rate for our multi-label setting. We also report per-dataset attack performances Appendix E.2. These results yield four main observations. (i) Original LR and NN models yield the highest attack scores, underscoring their vulnerability when released without protection. (ii) Larger $\varepsilon$ and $b$ values correspond to higher data leakage, as expected. (iii) Averaged LR models are more vulnerable than vanilla ones, despite their similar predictive performance (see Appendix E.1). The variance reduction from cross-validation, while mitigating sample bias, amplifies differences across models and thus facilitates attacks. Notably, WB attacks on averaged models achieve nearly perfect classification. (iv) Attack scores increase with deeper levels of access, with SBB and WB achieving surprisingly high values in many cases. Although WB can theoretically be recovered from SBB in LR models, in practice this may require evaluation at specific or additional samples (see Appendix D); hence, SBB attacks sometimes underperform WB despite their theoretical equivalence. Somewhat unexpectedly, WB attacks on NNs achieve lower scores than BB attacks, illustrating the difficulty of extracting information from more complex models.

Table 1: Hamming scores (mean $\pm$ std) of adversarial multi-label classifiers. $^\star$Attacks use LR coefficients recovered from the TTs.

|  |  | 2-WBB | SBB | WB |
|---|---|---|---|---|
| LR | (vanilla) | $0.8178 \pm 0.0035$ | $0.9129 \pm 0.0089$ | $0.9330 \pm 0.0010$ |
|  | (averaged) | $0.9149 \pm 0.0058$ | $0.9910 \pm 0.0132$ | $0.9999 \pm 0.0000$ |
| LR-DP | ($\varepsilon = 0.1$) | $0.5314 \pm 0.0081$ | $0.5352 \pm 0.0064$ | $0.5088 \pm 0.0059$ |
|  | ($\varepsilon = 1$) | $0.5710 \pm 0.0074$ | $0.5808 \pm 0.0059$ | $0.5178 \pm 0.0107$ |
|  | ($\varepsilon = 10$) | $0.7163 \pm 0.0087$ | $0.7840 \pm 0.0140$ | $0.6403 \pm 0.0149$ |
|  | ($\varepsilon = 100$) | $0.7663 \pm 0.0060$ | $0.8610 \pm 0.0260$ | $0.8672 \pm 0.0076$ |
| TT-LR | ($b = 2$) | $0.6666 \pm 0.0025$ | $0.8231 \pm 0.0065$ | $0.7461 \pm 0.0025^\star$ |
|  | ($b = 6$) | $0.7535 \pm 0.0022$ | $0.8604 \pm 0.0066$ | $0.7979 \pm 0.0027^\star$ |
|  | ($b = 10$) | $0.7687 \pm 0.0020$ | $0.8710 \pm 0.0053$ | $0.8129 \pm 0.0021^\star$ |
| NN |  | $0.7375 \pm 0.0056$ | $0.8608 \pm 0.0240$ | $0.6336 \pm 0.0064$ |
| NN-DP | ($\varepsilon \approx 0.2$) | $0.5222 \pm 0.0050$ | $0.6186 \pm 0.0065$ | $0.5125 \pm 0.0059$ |
|  | ($\varepsilon \approx 1$) | $0.5085 \pm 0.0046$ | $0.6420 \pm 0.0043$ | $0.5033 \pm 0.0043$ |
|  | ($\varepsilon \approx 10$) | $0.5924 \pm 0.0055$ | $0.6659 \pm 0.0359$ | $0.5008 \pm 0.0045$ |
|  | ($\varepsilon = \infty$) | $0.6198 \pm 0.0036$ | $0.7370 \pm 0.0229$ | $0.6360 \pm 0.0054$ |
| TT-NN | ($b = 2$) | $0.5759 \pm 0.0055$ | $0.6184 \pm 0.0209$ | $0.5061 \pm 0.0053$ |
|  | ($b = 6$) | $0.6252 \pm 0.0051$ | $0.8267 \pm 0.0167$ | $0.5018 \pm 0.0063$ |
|  | ($b = 10$) | $0.6544 \pm 0.0057$ | $0.8215 \pm 0.0241$ | $0.5025 \pm 0.0063$ |

As expected, DP models with larger privacy budgets $\varepsilon$ yield higher attack scores. As shown in Appendix E.1, increasing $\varepsilon$ (and thus reducing noise) also improves utility: for the largest $\varepsilon$, balanced accuracy and AUC approach the non-DP levels—although attacks still perform worse than in the non-DP case, indicating that even negligible amounts of noise help protect privacy. For NN-DP models, however, good predictive performance is only recovered at $\varepsilon = \infty$, highlighting the substantial impact of DP-SGD on NN training.

For TT models, we observe that even with $b = 2$ the predictive accuracy is already close to that of the original models. Although larger $b$ values slightly improve performance, the gains are modest compared to the sharp increase in attack success. This highlights a key advantage of tensorization as a knowledge-distillation mechanism: it reconstructs an effective TT model from highly restricted information (discretized outputs) while preserving strong utility. In contrast, DP methods inject noise directly into the learning process, which leads to more substantial performance degradation. As shown in prior work (Jia et al., 2019; Yang et al., 2020; Ye et al., 2022), modifying outputs can mitigate membership inference; tensorization reinforces this by reintroducing obfuscation at the WB level, since TT parameters are derived from already-obfuscated evaluations.

This also explains why WB attacks on TT-LR models—based on reconstructing LR coefficients from TT evaluations (Appendix D)—do not outperform BB attacks on the original LRs: the TT cannot leak more information than what is visible through $b$-WBB access to the LR outputs. For TT-NN models, where NN parameters cannot be recovered, gauge randomization fully obfuscates WB access, yielding attack accuracies around 50%.

Overall, TT models provide the best privacy–utility trade-off: models with $b = 2$ offer strong privacy protection, comparable to DP solutions with small $\varepsilon$ but with substantially better performance. Moreover, the TT representation grants additional interpretability benefits, as described in Section 4.

### 3.4.1 EXAMPLE: CHO1 VS. CHO1 + KATO

As an illustrative case, we consider the extreme task of distinguishing models trained only on Cho1 (964 samples) from those trained on Cho1 plus the small Kato cohort (35 samples). This simulates a high-risk scenario where an adversary detects the inclusion of a very small subgroup, approaching individual membership inference. Table 2 shows attack accuracies for the Kato label. As expected, 2-WBB attacks are nearly random. In contrast, averaged LRs reach $\sim$92% detection under SBB and achieve perfect classification under WB. Notably, even vanilla LRs under WB access attain $\sim$71% accuracy. Attacks on NN models and their tensorizations are nearly random guessers across all access types, likely due to the higher complexity of NNs. This, in turn, highlights the extreme vulnerability of LR models, which are widely adopted in critical settings for their interpretability.

Table 2: Hamming scores (mean $\pm$ std) of adversarial classification of models trained on Cho1 or Cho1+Kato, evaluated on the Kato label. $^\star$Attacks use LR coefficients recovered from the TTs.

|  |  | 2-WBB | SBB | WB |
|---|---|---|---|---|
| LR | (vanilla) | $0.5981 \pm 0.0323$ | $0.6217 \pm 0.0627$ | $0.7141 \pm 0.0245$ |
|  | (averaged) | $0.5065 \pm 0.0228$ | $0.9182 \pm 0.1490$ | $1.0000 \pm 0.0000$ |
| TT-LR | $(b = 2)$ | $0.5375 \pm 0.0312$ | $0.5811 \pm 0.0496$ | $0.5521 \pm 0.0208^\star$ |
| NN |  | $0.5253 \pm 0.0796$ | $0.4641 \pm 0.0788$ | $0.5246 \pm 0.1041$ |
| TT-NN | $(b = 2)$ | $0.4779 \pm 0.0850$ | $0.4962 \pm 0.0751$ | $0.5152 \pm 0.0765$ |

For context, Appendix E.3 reports model performance on Kato. TT models show some degradation, especially in AUC, but this does not fully explain the attack results: the performance gap between training on Cho1 or Cho1+Kato is similar for TTs and LRs, and NNs achieve even higher accuracies while leaking no information.

Overall, these results show that even a 35-sample cohort can be reliably identified within a large dataset. Model averaging and WB access amplify leakage, while TT models remain robust and do not reveal the presence of Kato under any access type.

### 3.4.2 ATTACKING PUBLICLY AVAILABLE MODELS

We illustrate the risk of WB attacks on publicly available LR coefficients from LORIS: (i) those released in Chang et al. (2024), and (ii) coefficients we reconstructed from the online interface.[2] Although the interface returns rounded probabilities rather than exact scores, by approximately inverting the monotonic mapping created for LORIS (Fig. 3) we obtain usable coefficients (Appendix D).

Applying our WB attack, Table 3 shows that Cho1 is correctly identified as the training dataset in both cases, consistent with Chang et al. (2024). Recovered coefficients are noisier, assigning also high probability to Cho2, but Cho1 remains dominant. Since Cho1 and Cho2 are train/test splits of the same dataset, both drawn from the same patient cohort, this spurious assignment likely reflects shared data characteristics. These results demonstrate that even with noisy reconstructed coefficients, adversaries can still infer training data membership with high confidence, highlighting the privacy risks of releasing or exposing LR parameters.

Table 3: WB attack scores for LORIS coefficients, using (i) the released parameters (Chang et al., 2024) and (ii) coefficients reconstructed from the online interface.

|  | Cho1 | Cho2 | MSK1 | MSK2 | Shim | Kato |
|---|---|---|---|---|---|---|
| Released | **1.0000** | 0.0007 | 0.0001 | 0.0000 | 0.0003 | 0.0000 |
| Reconstructed | **0.9944** | 0.8138 | 0.0440 | 0.0173 | 0.0005 | 0.0007 |

## 4 INTERPRETABILITY WITH TENSOR TRAINS

Beyond privacy guarantees, interpretability is essential in clinical prediction. The utility of LORIS lies not only in its accuracy, but also in its ability to provide insights into relevant features and produce scores monotonically correlated with response probability. Here we show that TT models retain similar interpretability, leveraging efficient computation of marginal and conditional distributions.

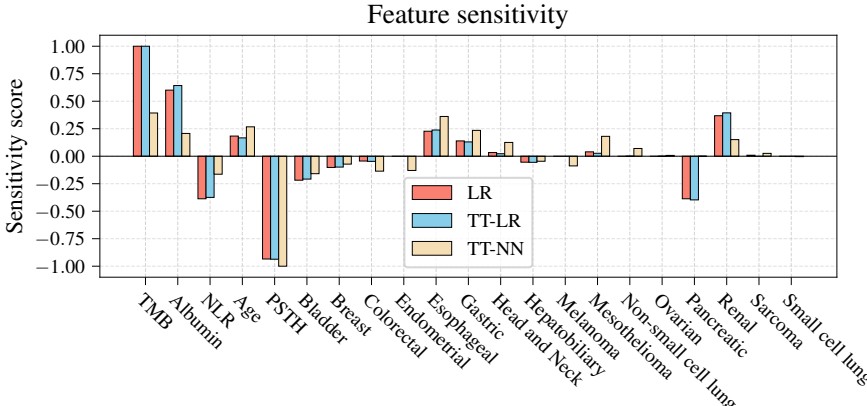

Figure 1: Feature sensitivity scores from LR and TT models (TT-LR and TT-NN, with $b = 6$). LR scores are coefficients, while TT scores are obtained via marginalization. All values are normalized by the maximum absolute score.

### 4.1 FEATURE SENSITIVITY

In LR, interpretability stems from coefficients, which quantify each feature's contribution through odds ratios. Since TTs approximate LRs, coefficients can in principle be recovered from TT outputs, but TTs also enable richer interpretability beyond linear models. Unlike LRs, where each feature has

---

[2]LORIS is available at: `https://loris.ccr.cancer.gov/`

a constant effect, TT sensitivities may vary with other features due to non-linearity. To emulate LR coefficients, we marginalize over all but one feature and the response, and measure how the predicted score changes under a unit increment of the selected feature. This procedure yields independent sensitivity scores that can be computed efficiently within the TT structure (Appendix A).

To evaluate this approach, we tensorized a vanilla LR trained on Cho1 and compared TT sensitivity scores with LR coefficients. We also included a tensorized NN model to assess how NN-based sensitivities compare to LR insights. As shown in Fig. 1, LR and TT-LR scores align almost perfectly after normalization by the maximum absolute value to remove scale differences. TT-NN yields similar relative patterns, albeit with larger scaling differences.. These results confirm that TTs preserve LR interpretability while extending the framework to more complex black-box models.

## 4.2 FEATURE SENSITIVITY BY CANCER TYPE

TTs also allow conditional analysis, enabling sensitivity computation for specific subgroups. Conditioning on cancer type produces smaller TT models that capture type-specific behaviors. Unlike the normalized comparison above, scores are directly comparable across cancer types since they are computed with the same method.

Figure 2 shows feature sensitivities for colorectal, endometrial, esophageal, and pancreatic cancers. While LR would provide identical scores across types, TTs reveal subtle variations. In particular, pancreatic cancer yields uniformly small sensitivities. This occurs because all pancreatic cancer patients in Cho1 are non-responders: the model achieves 100% accuracy simply by assigning very low response probabilities to all samples, independently of their features. Consequently, no feature appears relevant for prediction within this subgroup. These results highlight how TT interpretability can reveal subgroup-specific effects not captured by linear models.

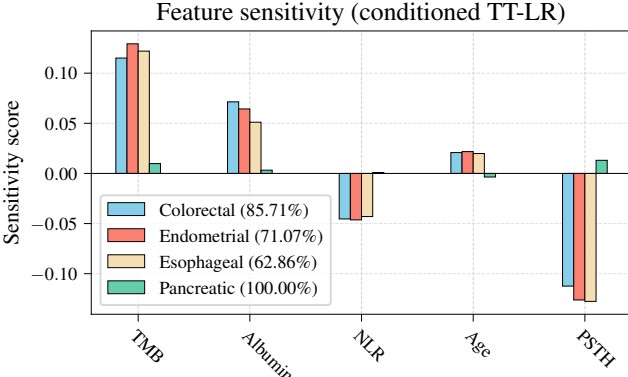

Figure 2: Feature sensitivities from conditioned TT-LR models ($b = 6$). The legend indicates cancer type and balanced accuracy of each conditioned TT on the corresponding data.

## 4.3 MONOTONICITY OF TT SCORES

A key property of LORIS scores, highlighted by Chang et al. (2024), is their monotonic relation with response probability. Although LR models are trained on binary labels, their scores align with mean response probabilities across patients sharing a given score. We verify this via bootstrapping to compute 95% confidence intervals for a vanilla LR model trained on Cho1. For comparison, we construct the same mapping for two tensorized LR models, using $b = 6$ and $b = 20$ bins for discretization.

Figure 3 shows the results. With $b = 6$, TT scores yield a lower slope, reflecting the discretization described in Section 3.2, which pushes the model toward more extreme values. Increasing the bin count improves the approximation, producing a mapping close to that of the LR model, though with potentially weaker privacy guarantees.

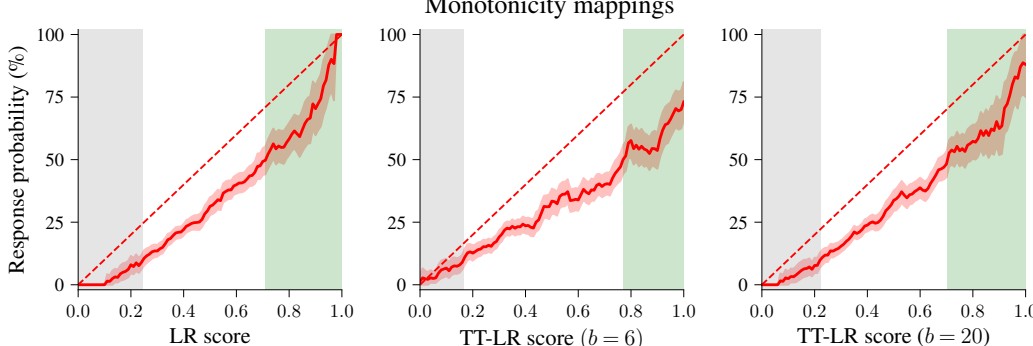

Figure 3: Monotonicity plots of LR and TT-LR models with different bin sizes. Shaded regions indicate participants with unlikely (gray, response probability $< 10\%$) or likely (green, response probability $> 50\%$) treatment response. From left to right, the limits of these regions are (0.25, 0.71), (0.17, 0.77), and (0.22, 0.70).

## 5  CONCLUSIONS AND DISCUSSION

In this work, we proposed tensorizing ML models into quantum-inspired TT representations as a mechanism to enhance privacy while preserving performance and interpretability. Through an empirical study of LORIS, we showed that models trained on small and sensitive datasets are highly vulnerable to training data leakage, underscoring the need for effective privatization. Our results further indicate that, although cross-validation is useful for model selection, averaging LR models for deployment should be avoided, as it amplifies privacy risks without offering meaningful accuracy gains. However, ensemble methods may still be privacy-preserving when the individual models and/or the aggregation method are private, as in PATE (Papernot et al., 2017). For linear models such as LR, where WB access can be reconstructed from SBB, releasing raw outputs without protection is particularly dangerous, as coefficients can be recovered to enable near-perfect identification of training data.

Our analysis of defense mechanisms reveals several key findings. For DP, our results confirm prior work (Ziller et al., 2024): only large $\varepsilon$ values are practical, while meaningful ones severely degrade accuracy. Tensorization, acting as a form of knowledge transfer, provides post-processing protection at all access levels. BB privacy arises from tensorizing obfuscated rather than raw scores—here achieved by discretizing outputs, though methods such as Yang et al. (2020); Ye et al. (2022) are equally applicable—while WB privacy follows from Pozas-Kerstjens et al. (2024).

Comparing TT-based protection with DP, we find similar privacy and performance for certain combinations of $b$ and $\varepsilon$, although a direct one-to-one comparison is not possible without DP-style output perturbation. Our results nevertheless suggest that the variability introduced by discretization plays a role analogous to noise injection. Thus, $b$ naturally acts as a privacy–utility knob. Notably, we show that even binary output access ($b = 2$) suffices to reconstruct TT models with accuracy close to the original, while offering privacy protection comparable to DP with small $\varepsilon$, yielding a more favorable privacy–utility trade-off than DP.

Beyond privacy, we showed that TTs recover LR interpretability while enabling richer analyses, including subgroup-specific effects. More importantly, TT interpretability extends naturally to tensorized NNs, suggesting that our approach can help "open the box" of otherwise opaque models. Thus, even when privacy is not the primary goal, tensorization provides a powerful framework for extracting insights from pre-trained models, reinforcing its value as a broadly applicable tool for both privacy and interpretability.

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

## REPRODUCIBILITY STATEMENT

We provide code to reproduce all experiments at `https://anonymous.4open.science/r/tts4privacy`. Our models are implemented with Scikit-Learn (Pedregosa et al., 2011), Diffprivlib (Dwork, 2006b), PyTorch (Paszke et al., 2019), Opacus (Yousefpour et al., 2022), and TensorKrowch (Pareja Monturiol et al., 2024). Details on datasets, preprocessing, hyperparameters, and training procedures are included in Section 3.3 and the appendices. All experiments were run on an Intel Xeon CPU E5-2620 v4 with 256 GB RAM and a single NVIDIA GeForce RTX 3090 GPU.

LLM USAGE STATEMENT

The authors used ChatGPT solely to improve the readability and language of the manuscript. All scientific content, including methods, results, and analysis, was developed by the authors. The authors reviewed and edited the text after using this tool and take full responsibility for the published content.

## A EFFICIENT COMPUTATIONS WITH TTs

A major advantage of tensor networks is their ability to represent high-order tensors using only a polynomial number of parameters. The TT representation of a tensor $T$ is given by

$$T(i_1, \ldots, i_N) = G_1(i_1) \cdots G_N(i_N), \tag{4}$$

requiring only $\mathcal{O}(Ndr^2)$ coefficients when all cores $G_n$ are $r \times r$ matrices, as opposed to the $d^N$ coefficients needed for a general tensor $T \in \mathbb{R}^{d^N}$. While compactness does not automatically imply fast computation, TTs are efficient to evaluate: computing $T(i_1, \ldots, i_N)$ scales polynomially in $N$, unlike higher-dimensional TNs where evaluation may require exponential time.

Beyond evaluating samples, TTs enable efficient marginalization. Suppose $T$ encodes a probability distribution via the Born rule, $p(i_1, \ldots, i_N) = |T(i_1, \ldots, i_N)|^2$. Computing the partition function,

$$Z = \sum_{i_1, \ldots, i_N} p(i_1, \ldots, i_N), \tag{5}$$

is generally exponential in $N$, but in TT form it reduces to polynomial time by contracting each core with itself:

$$H_n(\alpha_{n-1}, \beta_{n-1}, \alpha_n, \beta_n) = \sum_{i_n} G_n(\alpha_{n-1}, i_n, \alpha_n) \, G_n(\beta_{n-1}, i_n, \beta_n), \tag{6}$$

yielding $r^2 \times r^2$ matrices $H_n$. Multiplying all $H_n$ sequentially produces $Z$ efficiently.

A similar procedure yields marginals by contracting only the cores of marginalized features. For instance, for a 2-site TT

$$T(i, j) = G_1(i)G_2(j), \tag{7}$$

the marginal $p(i)$ is

$$p(i) = \sum_{\alpha, \beta} G_1(i, \alpha)G_1(i, \beta) \, H_2(\alpha, \beta), \tag{8}$$

showing that marginals correspond to duplicate TTs with some cores contracted.

TT representations also enable efficient computation of conditional models without retraining. To compute $p(i_1, \ldots, i_{n-1}, i_{n+1}, \ldots, i_N \mid i_n = \mathbf{i}_n)$, it suffices to absorb the fixed feature into its neighbor:

$$\widetilde{G}_{n-1}(i_{n-1}) = G_{n-1}(i_{n-1}) \, G_n(\mathbf{i}_n), \tag{9}$$

which defines a reduced, conditioned TT

$$\widetilde{T}(i_1, \ldots, i_{n-1}, i_{n+1}, \ldots, i_N) = G_1(i_1) \cdots \widetilde{G}_{n-1}(i_{n-1})G_{n+1}(i_{n+1}) \cdots G_N(i_N). \tag{10}$$

For further details on TTs and related tensor networks, see Cirac et al. (2021).

## B DATA STANDARDIZATION AND PARAMETER RESCALING

Before training on each dataset $D = \{(x_1^k, \ldots, x_n^k, y^k)\}_k$, input features $x_1, \ldots, x_n$ are standardized as

$$\tilde{x}_j^k = \frac{x_j^k - \mu_j}{\sigma_j}, \tag{11}$$

where $\mu_j$ and $\sigma_j$ denote the mean and standard deviation of feature $j$, respectively.

LR models are trained on these standardized inputs, but their parameters must be corrected in order to operate directly on raw features. Let $\tilde{\theta} = (\tilde{\mathbf{w}}, \tilde{b})$ be the parameters obtained after training, defining

$$\Phi_{\tilde{\theta}}(\mathbf{x}) = \text{sigmoid}(\tilde{\mathbf{w}}^{\mathsf{T}}\mathbf{x} + \tilde{b}), \quad \text{where} \quad \text{sigmoid}(z) = \frac{1}{1 + e^{-z}}. \tag{12}$$

The corrected parameters are $\theta = (\mathbf{w}, b)$ with

$$w_j = \frac{\tilde{w}_j}{\sigma_j}, \quad b = \tilde{b} - \sum_j \frac{\tilde{w}_j \mu_j}{\sigma_j}. \tag{13}$$

This transformation ensures that trained models can be applied directly to raw inputs without explicit feature standardization.

An analogous rescaling applies to TT models. Consider a tensorized model with parameters $\widetilde{\mathcal{W}}$,

$$\tilde{f}(x_1, \ldots, x_N) = \sum_{i_1, \ldots, i_N} \widetilde{\mathcal{W}}(i_1, \ldots, i_N) \, \phi_1(i_1, x_1) \cdots \phi_N(i_N, x_N), \tag{14}$$

where $\phi_n(\cdot, x) = [1, x]$ are polynomial embeddings (input dimension $d = 2$), and

$$\widetilde{\mathcal{W}}(i_1, \ldots, i_N) = \widetilde{G}_1(i_1) \cdots \widetilde{G}_N(i_N). \tag{15}$$

To compensate for feature standardization, we define a new coefficient tensor $\mathcal{W}$ from corrected cores $G_n$ such that

$$\begin{aligned} G_n(1) &= \widetilde{G}_n(1) - \frac{\mu_j}{\sigma_j} \widetilde{G}_n(2), \\ G_n(2) &= \frac{1}{\sigma_j} \widetilde{G}_n(2). \end{aligned} \tag{16}$$

The resulting TT parameters are thus expressed in terms of raw input features, analogous to the LR case.

## C    DATASETS DETAILS

In our study, we evaluate privacy vulnerabilities within the setting of Chang et al. (2024). The underlying task in that work is the prediction of treatment response in cancer patients receiving immune checkpoint blockade (ICB), using tumor mutational burden (TMB) together with additional clinical, genomic, and pathological variables. Although TMB has been proposed as a biomarker of ICB efficacy, it is not universally predictive across all cancers, motivating the development of multivariate models that combine TMB with other patient characteristics. Importantly, all patient cohorts considered originate from studies that pursue this same task of predicting binary ICB response.

The datasets used in this setting consist of multiple patient cohorts spanning 18 solid tumor types and up to 18 features, including tumor information, standard clinical variables, and blood-based markers. The represented cancer types include: non-small cell lung (NSCLC), renal, melanoma, head and neck, bladder, sarcoma, gastric, central nervous system (CNS), colorectal, endometrial, hepatobiliary, cervical (CLC), esophageal, pancreatic, mesothelioma, ovarian, breast, and cancers of unknown primary.

To establish a common modeling framework across heterogeneous cohorts, Chang et al. (2024) proposed a six-feature logistic regression model—the LORIS score—based on TMB, Previous Systematic Therapy History (PSTH), Albumin, Neutrophil-to-Lymphocyte Ratio (NLR), Age, and Cancer Type. Table 4 summarizes the cohorts and their main characteristics.

Table 4: Summary of the main dataset characteristics, including cohort size, cancer types, number of features provided, and original references.

| Dataset | Size | Cancer types | #Features | Reference |
|---------|------|--------------|-----------|-----------|
| Cho1 | 964 | 16 solid tumors | 18 | Chowell et al. (2022) |
| Cho2 | 515 | | | |
| MSK1 | 453 | 15 solid tumors | 13 | Chang et al. (2024) |
| MSK2 | 104 | CNS / Unkown primary | 12 | |
| Shim | 198 | NSCLC | 13 | Shim et al. (2020) |
| Kato | 35 | 8 rare tumors | 6 | Kato et al. (2020) |

# D    RECOVERING LR COEFFICIENTS FROM SBB ACCESS

Since logistic regression is linear in the log-odds space,

$$\text{logit}(\mathbf{x}) = \log \frac{p(y = 1 \mid \mathbf{x})}{p(y = 0 \mid \mathbf{x})} = \mathbf{w}^\mathsf{T}\mathbf{x} + b, \tag{17}$$

its parameters can be exactly recovered from model evaluations on carefully chosen inputs. If queries to the zero vector and one-hot vectors $\mathbf{e}_j$ are allowed, the intercept $b$ is simply the logit at the zero vector, and each coefficient $w_j$ is given by the difference between the logit at $\mathbf{e}_j$ and the intercept.

More generally, when queries are restricted to inputs with all features strictly positive (as in Section 3.4.2, when attacking LORIS through its web interface), $w_j$ can be recovered from two inputs $\mathbf{x}, \mathbf{x}'$ that differ only in feature $j$:

$$w_j = \frac{\text{logit}(\mathbf{x}) - \text{logit}(\mathbf{x}')}{x_j - x'_j}. \tag{18}$$

Once the weights are obtained, the intercept can be recovered from

$$b = \text{logit}(\mathbf{x}) - \mathbf{w}^\mathsf{T}\mathbf{x} \tag{19}$$

for any input $\mathbf{x}$.

# E    ADDITIONAL PRIVACY RESULTS

In this appendix we provide additional results supporting the conclusions of Section 3.4. Specifically, we report: (i) performance metrics of models trained on Cho1 (the largest dataset with 964 samples) and evaluated on all datasets; (ii) per-dataset attack accuracies for LR, NN, and TT models; (iii) performance of models trained on Cho1 versus Cho1+Kato; and (iv) attack results on models trained exclusively on individual public datasets.

## E.1    PERFORMANCE OF MODELS TRAINED ON CHO1

As an illustrative case, Fig. 4 shows the overall performance of models trained exclusively on Cho1, reporting balanced accuracies across all public datasets. Tensorization occasionally produces degenerate models with accuracies near 50%, which, although rare, can distort mean values. For this reason, we report median accuracies and AUC scores in the following tables, as they better capture typical behavior. Since the remaining distributions are approximately Gaussian and symmetric, median and mean coincide, making median values representative.

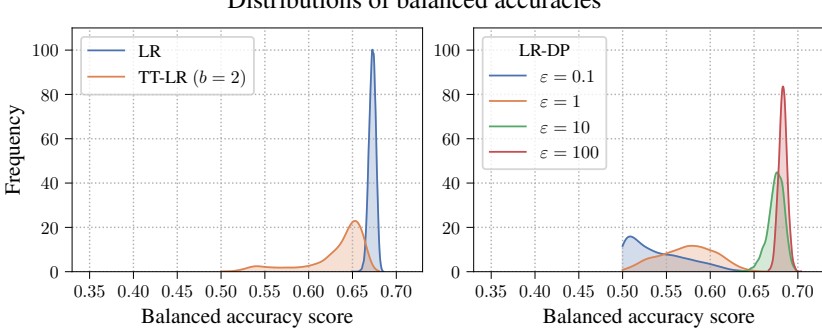

Figure 4: Balanced accuracy distributions of vanilla models trained on Cho1, evaluated on all samples from all datasets.

The right panel of Fig. 4 further shows how the performance of DP models improves with increasing $\varepsilon$, as the added noise decreases and the distribution converges to the narrow non-DP case.

Table 5 reports the median balanced accuracies across all datasets, and Table 6 presents the corresponding AUC scores. Together, these results reinforce the conclusions discussed in Section 3.4. Note that balanced accuracies use a threshold based on Youden's J statistic, as the standard 0.5 threshold produced irregular results for DP models.

Table 5: Median balanced accuracies of models trained on Cho1, evaluated on each dataset.

| | | Cho1 | Cho2 | MSK1 | MSK2 | Shim | Kato |
|---|---|---|---|---|---|---|---|
| LR | (vanilla) | 0.68 | 0.69 | 0.68 | 0.63 | 0.62 | 0.78 |
| | (averaged) | 0.68 | 0.69 | 0.69 | 0.63 | 0.62 | 0.78 |
| LR-DP | ($\varepsilon = 0.1$) | 0.52 | 0.53 | 0.53 | 0.56 | 0.53 | 0.58 |
| | ($\varepsilon = 1$) | 0.56 | 0.57 | 0.56 | 0.60 | 0.56 | 0.62 |
| | ($\varepsilon = 10$) | 0.67 | 0.68 | 0.66 | 0.63 | 0.61 | 0.70 |
| | ($\varepsilon = 100$) | 0.68 | 0.69 | 0.68 | 0.63 | 0.62 | 0.78 |
| TT-LR | ($b = 2$) | 0.66 | 0.66 | 0.65 | 0.63 | 0.61 | 0.70 |
| | ($b = 6$) | 0.67 | 0.67 | 0.67 | 0.63 | 0.62 | 0.72 |
| | ($b = 10$) | 0.67 | 0.68 | 0.67 | 0.63 | 0.62 | 0.73 |
| NN | | 0.72 | 0.69 | 0.64 | 0.63 | 0.62 | 0.80 |
| NN-DP | ($\varepsilon \approx 0.2$) | 0.58 | 0.59 | 0.53 | 0.63 | 0.57 | 0.62 |
| | ($\varepsilon \approx 1$) | 0.60 | 0.61 | 0.54 | 0.64 | 0.59 | 0.63 |
| | ($\varepsilon \approx 10$) | 0.61 | 0.62 | 0.56 | 0.64 | 0.60 | 0.65 |
| | ($\varepsilon = \infty$) | 0.68 | 0.68 | 0.63 | 0.66 | 0.62 | 0.75 |
| TT-NN | ($b = 2$) | 0.66 | 0.67 | 0.63 | 0.65 | 0.61 | 0.72 |
| | ($b = 6$) | 0.68 | 0.69 | 0.65 | 0.64 | 0.62 | 0.77 |
| | ($b = 10$) | 0.68 | 0.69 | 0.65 | 0.64 | 0.62 | 0.75 |

Table 6: Median AUC scores of models trained on Cho1, evaluated on each dataset.

|  |  | Cho1 | Cho2 | MSK1 | MSK2 | Shim | Kato |
|---|---|---|---|---|---|---|---|
| LR | (vanilla) | 0.74 | 0.75 | 0.70 | 0.63 | 0.60 | 0.75 |
|  | (averaged) | 0.74 | 0.75 | 0.70 | 0.63 | 0.60 | 0.71 |
| LR-DP | $(\varepsilon = 0.1)$ | 0.49 | 0.49 | 0.50 | 0.49 | 0.49 | 0.49 |
|  | $(\varepsilon = 1)$ | 0.56 | 0.57 | 0.55 | 0.55 | 0.54 | 0.51 |
|  | $(\varepsilon = 10)$ | 0.72 | 0.73 | 0.68 | 0.62 | 0.59 | 0.64 |
|  | $(\varepsilon = 100)$ | 0.74 | 0.75 | 0.70 | 0.63 | 0.60 | 0.75 |
| TT-LR | $(b = 2)$ | 0.69 | 0.69 | 0.67 | 0.62 | 0.60 | 0.62 |
|  | $(b = 6)$ | 0.72 | 0.72 | 0.69 | 0.63 | 0.60 | 0.65 |
|  | $(b = 10)$ | 0.72 | 0.72 | 0.69 | 0.63 | 0.60 | 0.65 |
| NN |  | 0.78 | 0.74 | 0.66 | 0.61 | 0.63 | 0.75 |
| NN-DP | $(\varepsilon \approx 0.2)$ | 0.59 | 0.59 | 0.49 | 0.60 | 0.53 | 0.55 |
|  | $(\varepsilon = 1)$ | 0.61 | 0.61 | 0.52 | 0.62 | 0.55 | 0.55 |
|  | $(\varepsilon = 10)$ | 0.65 | 0.63 | 0.56 | 0.64 | 0.57 | 0.57 |
|  | $(\varepsilon = \infty)$ | 0.73 | 0.72 | 0.65 | 0.63 | 0.62 | 0.73 |
| TT-NN | $(b = 2)$ | 0.70 | 0.69 | 0.65 | 0.62 | 0.60 | 0.67 |
|  | $(b = 6)$ | 0.73 | 0.74 | 0.67 | 0.62 | 0.62 | 0.73 |
|  | $(b = 10)$ | 0.73 | 0.73 | 0.67 | 0.62 | 0.62 | 0.71 |

### E.2 PER-DATASET ATTACK ACCURACIES

Table 7 reports per-dataset Hamming scores for LR, NN, and TT models, complementing the overall results in Section 3.4.

Table 7: Hamming scores (mean) of adversarial classification of vanilla models, separated by dataset.

|  |  | Cho1 | Cho2 | MSK1 | MSK2 | Shim | Kato |
|---|---|---|---|---|---|---|---|
| LR (vanilla) | 2-WBB | 0.9412 | 0.9358 | 0.9089 | 0.7237 | 0.7713 | 0.6259 |
|  | SBB | 0.9945 | 0.9782 | 0.9616 | 0.9626 | 0.9130 | 0.6675 |
|  | WB | 0.9982 | 0.9915 | 0.9678 | 0.9716 | 0.9152 | 0.7537 |
| TT-LR $(b = 2)$ | 2-WBB | 0.7530 | 0.6927 | 0.7100 | 0.6614 | 0.6357 | 0.5471 |
|  | SBB | 0.9343 | 0.8891 | 0.8740 | 0.8950 | 0.7517 | 0.5943 |
|  | WB$^\star$ | 0.8671 | 0.8064 | 0.7820 | 0.8517 | 0.5975 | 0.5722 |
| NN | 2-WBB | 0.9256 | 0.8826 | 0.7948 | 0.6104 | 0.6788 | 0.5329 |
|  | SBB | 0.7613 | 0.7091 | 0.6147 | 0.5507 | 0.5616 | 0.5128 |
|  | WB | 0.8335 | 0.7370 | 0.6545 | 0.5558 | 0.5111 | 0.5098 |
| TT-NN $(b = 2)$ | 2-WBB | 0.6429 | 0.6347 | 0.5832 | 0.5449 | 0.5450 | 0.5049 |
|  | SBB | 0.5796 | 0.5819 | 0.5445 | 0.5231 | 0.5224 | 0.4973 |
|  | WB | 0.5140 | 0.5133 | 0.5071 | 0.4972 | 0.5002 | 0.5049 |

### E.3 PERFORMANCE OF MODELS TRAINED ON CHO1 VS. CHO1+KATO

Table 8 reports the median balanced accuracies and AUC scores of models trained on Cho1 alone or on Cho1+Kato, evaluated on Kato. These findings support the results from Section 3.4.1.

Table 8: Median balanced accuracies and AUC scores of models trained on Cho1 or Cho1+Kato, evaluated on Kato.

| | | Bal. accuracy | | AUC | |
|---|---|---|---|---|---|
| | | Cho1 | Cho1+Kato | Cho1 | Cho1+Kato |
| LR | (vanilla) | 0.7833 | 0.8000 | 0.7533 | 0.7733 |
| | (averaged) | 0.7833 | 0.8167 | 0.7133 | 0.7800 |
| TT-LR | $(b=2)$ | 0.7167 | 0.7500 | 0.6167 | 0.6667 |
| NN | | 0.8000 | 0.8167 | 0.7467 | 0.8067 |
| TT-NN | $(b=2)$ | 0.7167 | 0.7333 | 0.6733 | 0.6733 |

### E.4 ATTACKS ON MODELS TRAINED ON INDIVIDUAL DATASETS

We also report attack performance on models trained exclusively on a single public dataset. This task is expected to be easier than identifying datasets within larger training sets.

Figure 5 compares accuracies in two scenarios. Rows indicate models trained on a given dataset (or on a larger set containing it), while columns correspond to evaluation on that dataset. As expected, accuracies are more uniform in the containment case (right), confirming that it is harder for the attacker than distinguishing models trained on distinct datasets (left).

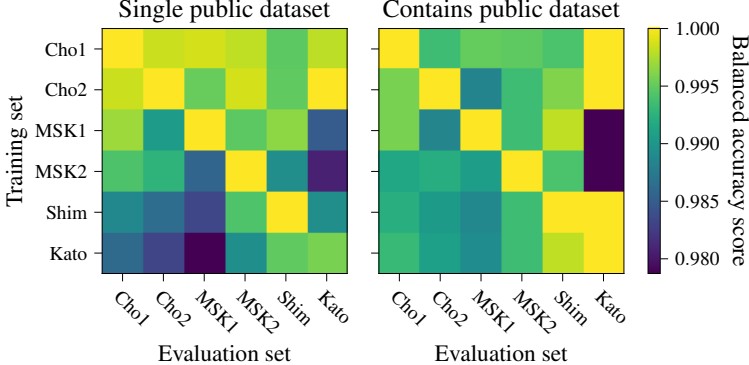

Figure 5: Median balanced accuracies of models evaluated across all datasets. Left: models trained on a single public dataset. Right: models trained on datasets containing each given public dataset.

Finally, Table 9 reports mean Hamming scores for attacks on models trained exclusively on individual datasets. While the relative performance across models is consistent with Table 1, the higher scores indicate that identifying training sets is even easier in this setting.

Table 9: Hamming scores (mean $\pm$ std) of attacks on models trained exclusively on a single dataset.
$^\star$Attacks use LR coefficients recovered from the TTs.

|  |  | 2-WBB | SBB | WB |
|---|---|---|---|---|
| LR | (vanilla) | $0.9139 \pm 0.0105$ | $0.9411 \pm 0.0152$ | $0.9592 \pm 0.0036$ |
|  | (averaged) | $0.9527 \pm 0.0187$ | $0.9819 \pm 0.0212$ | $0.9999 \pm 0.0002$ |
| LR-DP | $(\varepsilon = 0.1)$ | $0.5890 \pm 0.0345$ | $0.5956 \pm 0.0208$ | $0.2284 \pm 0.0460$ |
|  | $(\varepsilon = 1)$ | $0.5941 \pm 0.0283$ | $0.6031 \pm 0.0237$ | $0.2664 \pm 0.0616$ |
|  | $(\varepsilon = 10)$ | $0.7526 \pm 0.0233$ | $0.7856 \pm 0.0274$ | $0.6207 \pm 0.0429$ |
|  | $(\varepsilon = 100)$ | $0.8371 \pm 0.0242$ | $0.8638 \pm 0.0447$ | $0.8451 \pm 0.0150$ |
| TT-LR | $(b = 2)$ | $0.7164 \pm 0.0188$ | $0.8963 \pm 0.0151$ | $0.8318 \pm 0.0105^\star$ |
|  | $(b = 6)$ | $0.8517 \pm 0.0087$ | $0.9099 \pm 0.0168$ | $0.8678 \pm 0.0112^\star$ |
|  | $(b = 10)$ | $0.8662 \pm 0.0090$ | $0.9186 \pm 0.0115$ | $0.8809 \pm 0.0084^\star$ |
| NN |  | $0.7970 \pm 0.0261$ | $0.8770 \pm 0.0458$ | $0.6867 \pm 0.0187$ |
| NN-DP | $(\varepsilon \approx 0.2)$ | $0.3025 \pm 0.0432$ | $0.7455 \pm 0.0423$ | $0.5236 \pm 0.0178$ |
|  | $(\varepsilon \approx 1)$ | $0.2032 \pm 0.0408$ | $0.7176 \pm 0.0283$ | $0.5028 \pm 0.0148$ |
|  | $(\varepsilon \approx 10)$ | $0.5142 \pm 0.0389$ | $0.7236 \pm 0.0957$ | $0.4918 \pm 0.0183$ |
|  | $(\varepsilon = \infty)$ | $0.6258 \pm 0.0345$ | $0.7898 \pm 0.0337$ | $0.6421 \pm 0.0222$ |
| TT-NN | $(b = 2)$ | $0.5751 \pm 0.0403$ | $0.5973 \pm 0.0317$ | $0.5060 \pm 0.0230$ |
|  | $(b = 6)$ | $0.6213 \pm 0.0326$ | $0.8510 \pm 0.0208$ | $0.4843 \pm 0.0212$ |
|  | $(b = 10)$ | $0.7318 \pm 0.0384$ | $0.8685 \pm 0.0575$ | $0.5009 \pm 0.0190$ |

