# OpenReview forum: "Private and interpretable clinical prediction with quantum-inspired tensor train models"
_ICLR.cc/2026/Conference — Submitted to ICLR 2026_

### Official Review · Reviewer_9KFG · 2025-10-27

**Soundness:** 4
**Presentation:** 4
**Contribution:** 4
**Rating:** 8
**Confidence:** 2

**Summary:**

Authors first identifies vulnerabilities with privacy of LR models, and then propose a quantum-inspired defense using tensor train (TT) models to mitigate those vulnerabilities by tensorizing LR to obfuscate parameters while preserving accuracy and interpretability.

**Strengths:**

- Work tackles important issue of privacy of some of the widest used ML models in critical applications like medicine LR proposing a private method that remains interpretable and minimises leakage of personal data. Authors studied specifically membership inference attack based on shadow-model training.
- Authors reconstructed coefficients of the publically accessible model LORIS showing directly mentioned vulnerabilities

**Weaknesses:**

- The paper successfully demonstrates the TT defense on LR models, particularly LORIS, and notes that the approach only requires Black-Box (BB) access, making it generalizable to arbitrary models. This claim is a major potential contribution but currently lacks empirical validation beyond the linear model used.
- Preliminary theoretical framework would benefit the paper.

**Questions:**

The paper advises that averaging models for deployment should be avoided as common practices like cross-validation compromise privacy. Given that averaged models are often standard practice for stability, what specific alternative strategies do the authors recommend for the practical deployment of stable, yet privacy-preserving, models in sensitive clinical domains?

---

> ### Author Response · Authors · 2025-11-27
>
> We thank the reviewer for the time dedicated to evaluating our work and for the positive, valuable feedback that helped improve the paper. We have revised the paper accordingly, adding clarifications and incorporating new experiments; these required extra time, which explains the delay in our response. As the main concern shared across reviewers is now addressed in our general comment, we focus below on responding to the reviewer’s specific points.
>
> ---
>
> ### **Weaknesses**
>
> - ***“This claim is a major potential contribution but currently lacks empirical validation beyond the linear model used.”:*** We hope that the new experiments and the discussion provided in our general comment strengthen our generalizability claim.
>
> - ***“Preliminary theoretical framework would benefit the paper.”:*** In the revised version, we expanded the introduction and related work sections to better situate our method within prior literature (see, for instance, lines 64–70 or 110–124).
>
> ---
>
> ### **Questions**
>
> - ***“Given that averaged models are often standard practice for stability, what specific alternative strategies do the authors recommend for the practical deployment of stable, yet privacy-preserving, models in sensitive clinical domains?”:*** We thank the reviewer for raising this concern. As we clarify in the revised version—especially now that NN experiments are included—this averaging issue is specific to LRs (e.g., lines 512–514 in the discussion). It arises because Chang et al. (2024) average LORIS models by directly averaging parameters, which is feasible only due to linearity and may not extend generally to other models. Other cross-validation strategies remain compatible with privacy preservation, provided that each model is private and/or the aggregation step is also privatized. For example, Papernot et al. (2017) proposed PATE, a generalizable framework in which an ensemble of teacher models transfers knowledge to a student model via differentially private aggregation of ensemble predictions. This illustrates how stability and privacy can be jointly achieved without relying on parameter averaging.

---

### Official Review · Reviewer_oXJj · 2025-10-31

**Soundness:** 2
**Presentation:** 3
**Contribution:** 2
**Rating:** 2
**Confidence:** 4

**Summary:**

This paper shows that a linear regression (LR) models are vulnerable to membership inference attacks, and proposes a defense technique based on tensor trains (TT). The experimental results on LORIS, a publucly available immunotherapy response prediction model, shows that an attacker can accurately identify which datasets (from the nice candidates) were used to train the model. Then, the paper shows that applying the tensor train representation to the LR model makes this membership inference attack far less effecitve. Finally, the paper shows that the proposed TT approach still maintains the explainability of the original LR models.

**Strengths:**

The privacy concern in medical AI models is an important concern. The paper makes clear observations on the privacy risk of LR models and the potential privacy benefit of using TT. The experimental results suppor the main claims.

**Weaknesses:**

While demonstrating the privacy risk in a real-world immunotherapy prediction model is a notable contribution, the observation is not necessarily new or surprising from the technical point of view. Membership inference attacks have been shown for many types of models including more complex deep-learning models. It is relatively well-known that LR models are even more vulnerable given their simplicity and linear nature. In that sense, while the observation on the vulnerabiltiy adds another datapoint to what is known, it does not represent a new risk that was not known before. Also, the main privacy risk in medical models will be for individual personal records instead of entire datasets. In that sense, the study will be more compelling if the authors demonstrate attacks on individual records.

The proposed defense based on TT seems promising. However, the current security evaluation is only based on a specific type of an attack. Given that previous studies on other types of low-rank approximation (such as LoRA) did not provide sufficient privacy protection, it is unclear how robust the privacy protection from TT will truly be. Ideally, a privacy protection scheme needs to provide a mathematical guarantee. If not, the evaluation needs to be more comprehensive, including stronger learning based attacks as well as other privacy risks such as input reconstruction attacks, property inference attacks, model inversion attacks, etc.

There exist a large body of work on privacy risks in ML models. For related work, it will be helpful if the paper provides more comprehensive discusions. In particular, given that the paper proposes the TT-based defense, I would suggest including previous studies on low-rank approximation techniques and more explicitly state how this work is different.

**Questions:**

See above.

---

> ### Author Response · Authors · 2025-11-27
>
> We thank the reviewer for the time dedicated to evaluating our work and for the feedback provided. We have revised the paper accordingly, adding clarifications and incorporating new experiments; these required extra time, which explains the delay in our response. As the main concern shared across reviewers is now addressed in our general comment, we focus below on responding to the reviewer’s specific points.
>
> ---
>
> ### **Weaknesses**
>
> - ***“While demonstrating the privacy risk in a real-world immunotherapy prediction model is a notable contribution, the observation is not necessarily new or surprising from the technical point of view.”:*** We agree that many attacks have demonstrated the vulnerability of machine learning models. Although the susceptibility of LR models may be known, these models remain widely used and often unprotected—even in recent work—because of their interpretability and strong performance in data-limited settings such as clinical prediction. We aim to provide efficient, practical defenses that mitigate membership inference while preserving the performance and interpretability benefits of LRs. In the revised version, we also include NN-based experiments, further supporting our conclusions and concerns.
>
> - ***“Also, the main privacy risk in medical models will be for individual personal records instead of entire datasets.”:*** While membership inference is typically studied at the individual level, we believe our setting is similarly compelling. Instead of identifying a single patient, our attack targets small cohorts corresponding to specific studies or hospitals. In Sec. 3.4.1, we simplify the evaluation to detecting the Kato cohort of 35 samples within a training set of ~1000 samples—much closer to individual membership inference, as clarified in lines 352–353. Given the small cohort size, an adversary who identifies its inclusion could infer substantial information about the underlying individuals. Moreover, the structure of our attack closely parallels individual membership inference, and conducting such attacks would likely yield the same relative conclusions, albeit with lower absolute attack accuracies due to increased difficulty. Since Table 2 already shows that tensorization provides practical protection at the cohort level, it is very likely to offer stronger protection at the individual level.
>
> - ***“Given that previous studies on other types of low-rank approximation (such as LoRA) did not provide sufficient privacy protection, it is unclear how robust the privacy protection from TT will truly be. Ideally, a privacy protection scheme needs to provide a mathematical guarantee.”:*** We agree that our method does not provide formal black-box privacy guarantees, and that this previously limited the exposition. In the revised version (lines 64–70, 117–124, 518–523), we clarify the rationale behind our discretization strategy. Prior work—including Jia et al. (2019) and Yang et al. (2020)—shows that compressing the output space can impede membership inference without requiring DP guarantees, while Ye et al. (2022) present DP-based output discretization. Our approach thus builds on methods that deliver favorable privacy–utility trade-offs with or without DP. Crucially, the tensorization step is independent of the chosen obfuscation mechanism, and DP-based perturbations could be applied before tensorization. Our method is therefore not analogous to LoRA; a more accurate analogy is a knowledge-distillation process that learns only from obfuscated outputs. To clarify how our approach differs from previous work on low-rank decompositions, we have added a comment in lines 121–124. Finally, we note that the TT representation does provide formal white-box privacy guarantees, as shown by Pozas-Kerstjens et al. (2024), ensuring parameter-level obfuscation with only mild output perturbation.
>
> - ***“Given that the paper proposes the TT-based defense, I would suggest including previous studies on low-rank approximation techniques and more explicitly state how this work is different.”:*** In response, we expanded the introduction and related work sections to better situate our method within prior literature. We do not discuss low-rank approximation techniques in detail because our tensorization mechanism does not rely on those approaches, which are more specific to NNs. Instead, our method builds on three key ideas: (1) output obfuscation through discretization or noise injection to protect black-box access; (2) reconstruction of a new model (in TT form) solely from obfuscated outputs; and (3) white-box obfuscation arising from the the randomization of the gauge in the TT representation, an inherent property of tensor networks. However, we acknowledge that point (2) may be conceptually related to private low-rank approximation (Kapralov & Talwar, 2013). As noted earlier, we have added a clarification in lines 121–124.

---

### Official Review · Reviewer_xrcT · 2025-10-31

**Soundness:** 3
**Presentation:** 3
**Contribution:** 2
**Rating:** 4
**Confidence:** 4

**Summary:**

This paper studies privacy vulnerabilities in logistic regression models used for clinical prediction, focusing on the publicly available LORIS model for immunotherapy response. The authors design membership inference attacks under white-box and black-box access and show that cross-validation exacerbates leakage. As a defense, they propose tensorizing LR models using quantum-inspired tensor train representations, claiming that TT models retain interpretability and predictive performance while enhancing privacy, roughly comparable to Differential Privacy.

**Strengths:**

- The paper tackles a critical and timely issue at the intersection of clinical machine learning, privacy, and interpretability. Protecting sensitive medical data while maintaining transparent model behavior is a fundamental challenge for real-world deployment, and this work contributes a novel and practical perspective on it.
- The experimental setup is creative and insightful. The authors not only design realistic membership inference attacks but also demonstrate how such attacks can be applied to publicly available clinical models like LORIS. This empirical analysis illustrates the vulnerability of logistic regression models to privacy breaches, even under limited (black-box) access.
- The quantum-inspired tensor train representation provides an elegant defense mechanism that obfuscates model parameters while preserving the predictive accuracy and interpretability characteristic of logistic regression. The ability to maintain interpretability while improving privacy is a key contribution.

**Weaknesses:**

- The paper's primary contributions, including both the attack analysis and the proposed defense, are validated *exclusively* on Logistic Regression (LR) models. While the authors claim the tensorization method can be "applied to arbitrary models" because it only requires black-box access, this crucial claim is entirely unsubstantiated by the experiments. It remains unknown how effective this defense would be for more complex, non-linear models (e.g., deep neural networks).
- The Tensor Train (TT) defense mechanism proposed in the paper provides what is effectively empirical privacy; that is, it successfully thwarts the specific attacks designed by the authors. This is essentially a form of privacy by obfuscation, which lacks the rigorous, attack-agnostic mathematical guarantees of a framework like Differential Privacy (DP). It remains uncertain whether a more powerful adversary or an attack using different techniques could still break this defense.
- The paper's direct comparison between the TT method and DP is fundamentally inequitable because their privacy-utility mechanisms are non-equivalent. DP's trade-off is governed by the addition of calibrated noise, formally controlled by the $\epsilon$ parameter. The TT method's trade-off, however, stems from a different operation: information loss via discretization, controlled by the heuristic parameter $b$. This disparity makes a "fair" comparison inherently difficult, as it raises the question: what $\epsilon$ value in DP constitutes an equitable benchmark against the TT's $b=6$ setting? While the paper itself attempts to equate the two by suggesting $\epsilon=10$ offers a similar trade-off in empirical utility and robustness, this comparison is based on post-hoc performance observation. It misleadingly equates a formal, provable privacy guarantee with a heuristic, obfuscation-based one, simply because their empirical utility and robustness to a single attack happen to align at that specific setting.
- The paper's experimental comparison is largely limited to the original LR model and DP. This is insufficient to properly contextualize the contribution of the Tensor Train approach. The study would be significantly stronger if it benchmarked against other established privacy-preserving machine learning methods, even those mentioned in the related work , such as knowledge distillation or model pruning.

**Questions:**

- Could the authors provide preliminary evidence or at least a conceptual discussion of how the method would extend to non-linear or deep neural network architectures?
- The comparison between TT and DP relies on heuristic equivalence between the discretization parameter $b$ and the privacy budget $\epsilon$. Could the authors clarify how they define or justify this correspondence? Is there a principled way to calibrate $b$ so that it aligns with a given $\epsilon$, or are the two simply matched by post-hoc performance similarity?
- The study’s experimental comparisons are limited to LR and DP baselines. Have the authors considered benchmarking against other privacy-preserving strategies mentioned in the related work, such as pruning-based obfuscation or knowledge distillation?
- Table 1 shows that a WB attack on the TT cores results in random guessing, but an attack on reconstructed LR-TT coefficients is much more successful. Doesn't this imply that the function learned by the TT is still vulnerable to reconstruction and attack?

---

> ### Author Response · Authors · 2025-11-27
>
> We thank the reviewer for the time dedicated to evaluating our work and for the valuable feedback that helped improve the paper. We have revised the paper accordingly, adding clarifications and incorporating new experiments; these required extra time, which explains the delay in our response. As the main concern shared across reviewers is now addressed in our general comment, we focus below on responding to the reviewer’s specific points.
>
> ---
>
> ### **Weaknesses**
>
> - ***“While the authors claim the tensorization method can be ‘applied to arbitrary models’ because it only requires black-box access, this crucial claim is entirely unsubstantiated by the experiments.”:*** The new experiments and the clarifications above strengthen our generalizability claim. As noted, tensorizing deep NNs may yield poor approximations due to their higher expressive capacity. However, similar tensorization approaches using alternative tensor network structures—e.g., Tree TNs or loopy TNs—may increase expressiveness and improve approximation quality for more complex models.
>
> - ***“The Tensor Train (TT) defense mechanism provides what is effectively empirical privacy [...]”:*** We agree that our method does not offer formal black-box privacy guarantees, and that this limitation made the previous exposition insufficiently substantiated. In the revised version (lines 64–70, 117–124, 518–523), we clarify the basis for our discretization strategy. Prior works such as Jia et al. (2019) and Yang et al. (2020) show that compressing the output space can hinder membership inference without requiring DP guarantees, using noise injection to induce misclassification by attackers. Similarly, Ye et al. (2022) propose DP-based output discretization. Our approach thus builds on methods that achieve strong privacy–utility trade-offs with or without DP guarantees. Importantly, we emphasize that the tensorization step is independent of the chosen obfuscation method, and DP-based output perturbation could be applied prior to tensorization. However, our goal in this work is to highlight practical, efficient post-processing methods for privacy and interpretability. Finally, we note that our TT approach does provide formal white-box privacy guarantees as proved by Pozas-Kerstjens et al. (2024), ensuring that parameters are inherently obfuscated and only mild output obfuscation is needed.
>
> - ***“The paper’s direct comparison between the TT method and DP is fundamentally inequitable because their privacy–utility mechanisms are non-equivalent.”:*** We thank the reviewer for pointing this out. In the revised paper, we have removed the post-hoc identification between $\varepsilon$ and $b$ and now treat both methods independently (see discussion, lines 524–530). We also include additional $b$ values (2, 6, 10) to better illustrate its effect (Sec. 3.4). Two observations follow:
> 1. The parameter $b$ acts as an intuitive privacy–utility knob, arguably more interpretable than $\varepsilon$, as it directly controls how much information from the original model is retained.
> 2. Although selecting $b$ post-hoc may seem unfair, a similar issue exists with DP: practical values of $\varepsilon$ that preserve utility are often extremely large, yet still prevent membership inference or reconstruction attacks in practice (e.g., Ziller et al., 2024). DP methods also typically incur significant computational overhead and require task-specific tuning. Given our emphasis on practical, efficient post-processing, we believe our approach offers a broadly applicable alternative, while acknowledging that integrating DP guarantees would be valuable future work.
>
> - ***“The paper's experimental comparison is largely limited to the original LR model and DP.”:*** The new NN experiments—both with and without DP—make the experimental evaluation more complete. Although comparisons with pruning, knowledge distillation, or other non-DP methods could offer additional insights, establishing fair baselines across such heterogeneous techniques is difficult. For example, pruning is NN-specific and cannot be applied to LRs. Knowledge distillation was tested in preliminary experiments, but training new LRs on discretized LR outputs led to identical attack performance, likely because the simplicity of LRs causes the distilled models to converge to the same parameter regimes. For these reasons, we focus on DP-based baselines, which offer the most consistent and fair point of comparison.

---

> ### Author Response · Authors · 2025-11-27
>
> ### **Questions**
>
> - ***“Could the authors provide preliminary evidence or at least a conceptual discussion of how the method would extend to non-linear or deep neural network architectures?”:*** As noted earlier, tensorization is architecture-agnostic: it constructs a TT function that approximates a model from evaluations on a small sample set. Thus, the method applies equally to LRs and NNs, with the only limitation being the expressive capacity of TTs, which may not approximate highly complex models well. For further reference, we direct the referee to Pareja Monturiol et al. (2025) and the works cited therein.
>
> - ***“The comparison between TT and DP relies on heuristic equivalence between the discretization parameter $b$ and the privacy budget $\varepsilon#. Could the authors clarify how they define or justify this correspondence? ”:*** The previous correspondence was purely post-hoc. In the revised version, we avoid making such equivalences and treat TT-based discretization and DP independently, discussing the privacy–utility trade-offs of each method separately. See, for example, the analysis in lines 324–346.
>
> - ***“Have the authors considered benchmarking against other privacy-preserving strategies mentioned in the related work, such as pruning-based obfuscation or knowledge distillation?”:*** As discussed above, we considered such methods but focused on DP for clarity and fairness.
>
> - ***“Table 1 shows that a WB attack on the TT cores results in random guessing, but an attack on reconstructed LR-TT coefficients is much more successful. Doesn't this imply that the function learned by the TT is still vulnerable to reconstruction and attack?”:*** For tensorized LRs, an attacker can reconstruct LR coefficients from the TT representation, making attacks stronger than what the TT white-box attack alone suggests. This limitation is specific to linear models: unlike NNs, LR parameters can be reconstructed from function values. To clarify this, Tables 1, 2, and 9 now report—for TT-LRs—the WB attack accuracies against the reconstructed LR coefficients, with additional explanation provided in lines 257–260. In contrast, for NNs, TT representations do not allow parameter reconstruction, and WB attacks on TT-NN models behave as random guessers.

---

### Official Review · Reviewer_7njC · 2025-11-05

**Soundness:** 2
**Presentation:** 2
**Contribution:** 2
**Rating:** 2
**Confidence:** 4

**Summary:**

This work uses quantum-inspired “tensor train” model representations to produce black-box tensorized representations of logistic regression (LR) models. The work is motivated by the tension between the transparency and the attack vulnerability of simple ML models used within medical/clinical decision making contexts. The authors:
* provide empirical evidence of logistic regression model vulnerabilities to adversarial attacks and access,
* demonstrate dataset leakage by an LR-based clinical model, LORIS (logistic regression-based immunotherapy response score),
* propose a defense training process using tensor train models that maintains accuracy and LR interpretability while decreasing attack vulnerability in the context of LORIS.

**Strengths:**

* Sec 3.3 Experimental Setup provides a detailed overview of the datasets, models, and implementation details. Suggested improvement under “Weaknesses”.
* Feature sensitivity analysis is interesting and supports model interpretability, which aligns with the problem motivation and justification for focusing on LR models within this work.
* Preliminary exploration seems potentially promising. (Reviewer believes more is needed, please refer to `Weaknesses`.)

**Weaknesses:**

* At a high-level, the problem this work centers upon seems important for the application area in question (logistic regression for immunotherapy response prediction). However, it’s unclear how broadly applicable the proposed approach (quantum-inspired defense using tensor train models) is for other tasks or domains. The paper currently reads, at least to this reviewer, as being better suited for a more targeted audience or venue than the broader ML community reflected at ICLR.
    * The scoping centers largely on LORIS, “a publicly available LR model for immunotherapy response prediction”, which is a very specific application of logistic regression models (which themselves are a very specific, simple, and classical ML model). This results in the paper’s findings being very tightly scoped with limited broader applicability (even if the proposed method may be more generally useful).
    * If the authors believe the scope of this work aligns with the ICLR community, a clearer introduction, background, and motivation of the setting is needed.
* In response to the last line of the abstract: `Although demonstrated on LORIS, our approach generalizes broadly, positioning TT models as a practical foundation for private, interpretable, and effective clinical prediction.`
    * While this may be the case, we need to have some evidence of the broad generalizability and applicability of the method for this claim to hold.
* Sec 3.3.1 Datasets would benefit from a table to provide an outline the datasets used, both for greater ease of understanding for the reader. Additional details on these datasets would be helpful to contextualize the tasks, so that these are not simply shorthand identifiers. In other words, why do these tasks matter, and what do they map to (e.g. `clinical, pathological, and genomic features with a binary treatment-response label`)? This would also help build greater intuition around the vulnerabilities of LORIS and the risks for applications in clinical settings to motivate employing a method as described in this work.
* Empirical results (in 3.4 and tables) are hard to follow. Suggest reorganizing to clarify the evaluation setup, individual tasks, results, and discussion.
* More discussion around the choice of metrics used to assess the adversarial attacks and their definition(s) is needed.

**Questions:**

* In terms of metrics, attack success rate is an important metric in evaluating adversarial attacks. Is this something you explored in the tasks evaluated here?
    * Additionally, adversarial robustness score would be useful, particularly to demonstrate the difference between LORIS and the increased robustness of using TT models.
* Under what basis can you support the claim of `Although demonstrated on LORIS, our approach generalizes broadly, positioning TT models as a practical foundation for private, interpretable, and effective clinical prediction.`? Demonstrating broader applicability would add more value and substance to this work.

---

> ### Author Response · Authors · 2025-11-27
>
> We thank the reviewer for the time dedicated to evaluating our work and for the detailed feedback. We have revised the paper accordingly, adding clarifications and incorporating new experiments; these required extra time, which explains the delay in our response. As the main concern shared across reviewers is now addressed in our general comment, we focus below on responding to the reviewer’s specific points.
>
> ---
>
> ### **Weaknesses**
>
> - ***"It’s unclear how broadly applicable the proposed approach is for other tasks or domains":*** We hope the discussion in our general comment clarifies the broad applicability of our approach.
>
> - ***"This results in the paper’s findings being very tightly scoped with limited broader applicability":*** We believe the new NN-based experiments strengthen the case for generalizability. Additionally, Pareja Monturiol et al. (2025) applied tensorization to larger NN architectures, further supporting this claim.
>
> - ***“Datasets would benefit from a table to provide an outline the datasets used”:*** We agree that a more detailed dataset summary would benefit Sec. 3.3.1. In the revised version, we include additional details on the task studied by Chang et al. and a brief table summarizing the datasets. For comprehensive information, we continue to refer readers to the original source, which provides a thorough description of the patient cohorts, the medical features considered, and their associated analyses.
>
> - ***“Empirical results (in Sec. 3.4 and tables) are hard to follow.”:*** In the revised version, we have streamlined the presentation of results. Although we added NN-based experiments, we reduced the number of DP models shown, simplified the vanilla/averaged distinction to only LRs, and replaced WB attack accuracies for TT-LRs with the (previously denoted) LR-TT parameters (explained in lines 257–260 and in the captions of Tables 1, 2, and 9). We hope these changes improve readability.
>
> ---
>
> ### **Questions**
>
> - ***“More discussion around the choice of metrics used [...] / Attack success rate is an important metric [...].”:*** We use Hamming accuracy, a standard metric in multi-label classification, because our attacker predicts independently for each public dataset whether it was included in the training set. Thus, Hamming accuracy is a natural analogue of attack success rate in our multi-label setting. We added a clarification on this in line 288.
>
> - ***“Additionally, adversarial robustness score would be useful.”:*** While we agree that exploring robustness benefits of TTs is an interesting direction, it is outside our scope. We now cite Mossi et al. (2025) for reference in line 132.

---

### Author Response · Authors · 2025-11-27
**General Response and List of Changes**

We thank all reviewers for their thorough evaluation and the valuable feedback provided. In response, we have revised the paper to address the main concerns, adding clarifications throughout and incorporating new experiments. These additional experiments required extra time, which explains the delay in our response, for which we apologize.

A major concern raised across reviews was that our claim of generality lacked sufficient support. Because this point appeared repeatedly, we address it here as a general comment. We also summarize the main changes introduced in the revised version of the paper.

---

### **Generality of our approach**

The method used to transform pre-trained models into TT form builds upon Pareja Monturiol et al. (2025), where the authors provide multiple tensorization examples, from simple functions to larger NN models such as image and audio classifiers. In the previous version of our paper, we referred to this work for NN examples but did not include them ourselves, as our focus was on LORIS. To better emphasize the methodological contribution rather than the specific application setting, we now include NN-based experiments—specifically, shallow MLPs also considered by Chang et al. (2024). Additionally, to remove the dependence of our exposition on the choice of the number of bins $b$ and to improve the analysis of its effect, we have incorporated experiments covering a broader range of $b$ values (specifically, $b = 2, 6, 10$).

To reduce computational cost, we simplified the task by removing the last 3 of the 9 public datasets originally used. Overall, results remain consistent, with a slight increase in attack accuracy likely due to this simplification. The new NN results confirm the conclusions of the previous version and further highlight the benefits of tensorization, which achieves favorable privacy–utility trade-offs while improving interpretability for both LRs and NNs. All tables and figures have been updated accordingly, and references and clarifications regarding the NN models are now included throughout the paper. Finally, the inclusion of NN models in our experiments also strengthens the interpretability results, as Figure 1 now includes sensitivities obtained from tensorized NNs. These results show that our method not only preserves the insights provided by transparent models such as LRs, but also extends interpretability to more complex architectures.

Regarding generality, we emphasize that the tensorization mechanism is architecture-agnostic and applicable to different model classes. It is also independent of the output obfuscation strategy: while we use discretization as a simple method to compress the output space, other noise-based approaches with DP guarantees could be applied prior to tensorization. These clarifications are explicitly made in the revised manuscript (e.g., lines 64–70, 117–124, 518–523). Nevertheless, we refrain from claiming that *any model* can be tensorized with comparable success, since approximation accuracy ultimately depends on the expressive power of the TT representation, which may pose limitations for highly complex models.

Despite this limitation, we believe that the setting and tasks we study are broadly representative. This is underscored by LORIS, recently introduced by Chang et al. (2024), where the authors compare a wide range of state-of-the-art models and find that logistic regression achieves the best performance while providing strong interpretability. Accordingly, although our work is framed around clinical prediction, it addresses a broad class of problems characterized by limited data availability and strong privacy and interpretability requirements. These scenarios include domains where TT models have demonstrated strong performance: interpretability in cybersecurity (Aizpurua et al., 2024), NLP (Tangpanitanon et al., 2022), and financial forecasting (arXiv:2105.04983), as well as state-of-the-art results in tabular anomaly detection (Wang et al., 2020) and proton collision event detection at the LHC (arXiv:2506.00102).

For these reasons, we believe our approach is highly generalizable and relevant to a broad audience, making ICLR an appropriate venue.

---

> ### Author Response · Authors · 2025-11-27
>
> ### **Changes to the revised version**
>
> - **Additional experiments:** We now include experiments with NN models, their DP variants, and their tensorizations. Tables 1, 2, 5, 6, 7, 8, and 9, as well as Figure 1, have been updated accordingly.
> - **Simplified adversarial setting:** To reduce computational cost, we removed the Vang, Ravi, and Pradat datasets. All conclusions remain unchanged. Tables 3, 5, 6, and 7, and Figure 5, reflect this update.
> - **Revised DP baselines:** We now focus on $\varepsilon = 0.1, 1, 10, 100$, since $\varepsilon = 0.01$ and $\infty$ yielded identical results to $\varepsilon = 0.1$ and $100$, respectively. For comparison, we include DP-trained NN models with $\varepsilon = 0.2, 1, 10, \infty$. Additionally, we improved our analysis of TT models by incorporating more bin sizes ($b=2, 6, 10$). Tables 1, 5, 6, and 9, and Figure 4, have been updated accordingly.
> - **Averaged vs. vanilla models:** We restrict this distinction to simple LRs, as all other model classes exhibited analogous behavior. Tables 1, 2, 5, 6, 8, and 9 have been updated.
> - **New Appendix C:** We added Appendix C with further details on the dataset and the task addressed by Chang et al., including a new Table 4 summarizing dataset characteristics.
> - **Expanded background:** Background has been improved in the abstract, introduction, and related work sections.
> - **Revised Section 3:** Section 3—particularly its introduction (lines 160–167), Sec. 3.3.2, and Sec. 3.4—has been fully revised to incorporate the new NN-based models and updated privacy experiments.
> - **Interpretability analysis:** Section 4.1 now includes interpretability results for tensorized NNs, and Figure 1 has been updated.
> - **Updated conclusions and discussion:** Lines 518–530 have been revised to reflect the new results.

---

### Meta-Review · Area_Chair_UiNc · 2025-12-26

**Summary:**

This paper investigates privacy risks in machine learning models (i.e., logistic regression) used in medical application settings. It focuses mostly on a public model called LORIS, which predicts how patients might respond to immunotherapy (i.e., cancer treatment). The authors show that an attack could figure out which patient data was used to train these model, particularly if they have full access to the model's details (i.e., a white-box attack). To protect medical models against this attack, the authors propose a technique inspired by tensor train or TT representation, which scrambles the model's parameters to make it harder to reverse-engineer while keeping the predictions accurate and understandable.

The reviewers think the paper focused on one narrow example (this LORIS model and logistic regression). The authors mostly just test their techniques without mathmetic theory or new ideas that apply to more types of models, like a deeper neural network. Reviewers had very different opinions: one really liked it, but most thought it was not strong enough. They felt it lacked new ideas, was not broad enough, and the claims were a bit overstated.

**Reviewer Concerns:**

Main Concerns from Most Reviewers:

1. Too Narrow Focus (One Model & One Use Case). The entire paper is built around logistic regression and one specific cancer prediction application (LORIS). Although the authors said this example represents bigger problems, but they did not prove it applies widely. Even the extra tests on more complex AI models (i.e., deeper neural networks) are limited.

2. Overstating How Well It Works for Any Model. The authors claimed their TT method works for "any" AI model, but almost all the proof was just on logistic regression. Tests on other models (i.e., deeper neural networks) were weak, and the authors admitted it might not work well for complicated ones.

3. Not Real Privacy Protection. The TT method helps hide the model from specific attacks, but it did not offer any strong mathematical guarantees like Differential Privacy (DP). Although the paper compared TT to DP, but reviewers said the comparison was not fair because the two work differently.

4. Not Enough Testing Against Different Attacks. The privacy tests only check one type of attack (figuring out whole datasets used in training) under specific conditions. Reviewers pointed out missing tests for: Attacks on individual patients, Stronger or smarter attacks, Other privacy methods (like compressing the model).

**Although the authors made some changes after the first reviews, but most reviewers did not buy.**


Review-Specific Questions:

1. Several reviewers (7njC, oXJj, q1SR) said the paper feels like a narrow case study for cancer researchers, not a big step forward in general AI.

2. One reviewer (xrcT) strongly disliked the TT vs. DP comparison, saying it is not fair and could mislead people about how safe TT is.

3. Another (oXJj) noted that logistic regression models being easy to attack is already well-known, so this is not new.

**I do not think the authors addressed the questions from reviewer oXJj well**. Future revisions should be performed based on the questions of reviewer oXJj.

**Reviewer Scores:**

I think the scores from reviewers are reasonable.

---

### Decision · Program_Chairs · 2026-01-26

Reject